# Novel Size-Variable Dedicated Rodent Oxygenator for ECLS Animal Models—Introduction of the “RatOx” Oxygenator and Preliminary In Vitro Results

**DOI:** 10.3390/mi14040800

**Published:** 2023-03-31

**Authors:** Lasse J. Strudthoff, Jannis Focke, Felix Hesselmann, Andreas Kaesler, Ana Martins Costa, Peter C. Schlanstein, Thomas Schmitz-Rode, Ulrich Steinseifer, Niklas B. Steuer, Bettina Wiegmann, Jutta Arens, Sebastian V. Jansen

**Affiliations:** 1Institute of Applied Medical Engineering, Department of Cardiovascular Engineering, Medical Faculty, RWTH Aachen University, 52074 Aachen, Germany; focke@ame.rwth-aachen.de (J.F.); felix.hesselmann@rwth-aachen.de (F.H.); andreas.kaesler@rwth-aachen.de (A.K.); schlanstein@ame.rwth-aachen.de (P.C.S.); steinseifer@ame.rwth-aachen.de (U.S.); steuer@ame.rwth-aachen.de (N.B.S.); j.arens@utwente.nl (J.A.); jansen@ame.rwth-aachen.de (S.V.J.); 2Department of Biomechanical Engineering, Faculty of Engineering Technologies, University of Twente, 7522 LW Enschede, The Netherlands; a.f.martinscosta@utwente.nl; 3Institute of Applied Medical Engineering, Medical Faculty, RWTH Aachen University, 52074 Aachen, Germany; smiro@ame.rwth-aachen.de; 4Department for Cardiothoracic, Transplantation and Vascular Surgery, Hannover Medical School, 30625 Hanover, Germany; wiegmann.bettina@mh-hannover.de; 5Lower Saxony Center for Biomedical Engineering, Implant Research and Development (NIFE), 30625 Hanover, Germany; 6German Center for Lung Research (DLZ), 30625 Hanover, Germany

**Keywords:** ECLS, ECMO, animal model, rodent, rat, mouse, hamster, CPB, in vivo, oxygenator

## Abstract

The overall survival rate of extracorporeal life support (ECLS) remains at 60%. Research and development has been slow, in part due to the lack of sophisticated experimental models. This publication introduces a dedicated rodent oxygenator (“RatOx”) and presents preliminary in vitro classification tests. The RatOx has an adaptable fiber module size for various rodent models. Gas transfer performances over the fiber module for different blood flows and fiber module sizes were tested according to DIN EN ISO 7199. At the maximum possible amount of effective fiber surface area and a blood flow of 100 mL/min, the oxygenator performance was tested to a maximum of 6.27 mL O_2_/min and 8.2 mL CO_2_/min, respectively. The priming volume for the largest fiber module is 5.4 mL, while the smallest possible configuration with a single fiber mat layer has a priming volume of 1.1 mL. The novel RatOx ECLS system has been evaluated in vitro and has demonstrated a high degree of compliance with all pre-defined functional criteria for rodent-sized animal models. We intend for the RatOx to become a standard testing platform for scientific studies on ECLS therapy and technology.

## 1. Introduction

Patients suffering from critical cardiac and/or respiratory dysfunction of any cause can be bridged to treatment or recovery by extracorporeal circulation that can comprise an artificial heart, lungs and a heat source. Extracorporeal life support (ECLS) is a high-risk procedure and is considered a late therapy escalation or even a last-resort measure [1]. Technology, knowledge and experience in the use of ECLS are evolving rapidly, and as a result, an increasing number of indications are being treated with ECLS, and more patients are considered eligible after weighing risks against potential benefits. The overall increase in ECLS deployments can be seen in the figures regularly published by the Extracorporeal Life Support Organization (ELSO); these figures reflect the worldwide deployment of ECLS by centers registered with the ELSO. Irrespective of the various medical indications, more than 176,000 patients have received ECLS between 1990 and today. After an initial surge in deployments following the September 2008 H1N1 pandemic, ECLS was also an important therapy during the COVID-19 pandemic [2]. This increases the relevance of ECLS as a therapy and signifies the demand for expedited ECLS research models, especially as ECLS still suffers from many complications [3].

Research and development of ECLS technology and therapy has been ongoing since its introduction in the 1970′s. Nonetheless, survival rates remain relatively stable at only 60% for patients suffering from respiratory failure and 30–50% for patients suffering from cardiorespiratory failure [3,4]. There have been no major breakthroughs since the introduction of hybrid dense polymethylpentene membranes (PMP). In the context of ECLS, there are multiple knowledge gaps and, thus, barriers to innovation [5], listed here from three perspectives: (1) Clinical research: ECLS is a relatively low-volume therapy. From 2017 to 2021, only 85,318 cases have been recorded by ELSO, including a plausible sudden increase due to COVID-19 patients [3]; this means smaller patient cohorts for any clinical trial and a non-economic cost–benefit ratio for research in this area. In addition, the patient population is highly heterogeneous, and the underlying causes for a patient to require ECLS are manifold. Hence, forming patient collectives with similar baseline characteristics is often unfeasible. (Multiple) blinding is practically impossible due to immediate unblinding. In the majority of cases, ECLS is a high-risk, last-resort therapeutic measure; patient selection for the merit of a study is ethically difficult. As only critical patients with primary pathologies are treated with ECLS, the actual cause of death, despite or due to ECLS therapy, is often difficult to isolate. (2) Modeling: ECLS is a very problematic therapy in terms of hemocompatibility and blood damage. Extremely large artificial surfaces in the extracorporeal systems are coupled with non-physiological flow fields, which have not yet been deciphered in depth. Testing this hemocompatibility usually requires animal experiments. Typically, porcine models are used for preclinical testing [6]. These studies are cumbersome, expensive, ethically challenging, and require extensive planning. Current in vitro test methods are not sufficiently conclusive. Further, many complications of ECLS occur days or weeks after therapy initiation. This implies that it is often difficult or impossible to model the processes that lead to a particular complication. (3) Technology: Different commercially available ECLS oxygenator designs exhibit different inflammatory and hemocompatibility behaviors for similar therapeutic applications [7,8] with limited knowledge of the causes, pathways, magnitude, and impact. Further impeding research on the key component of the technology, the gas-exchanging hollow-fiber membranes, is the unbalanced market for these fibers; the technology leader (Membrana GmbH, 3M, Wuppertal, Germany) has a very large market share in a very small market: Economically, this provides little incentive for innovation.

To overcome the obstacles listed above and to study both therapy and technology, researchers have begun to use rodent models in their ECLS studies, as listed in Table 1. Cardiopulmonary bypass (CPB) models are not included in this table. Although ECLS originates from CPB and shares several features, the duration of use is in the range of hours for CPB and days to weeks for ECLS, to name the primary distinguishing characteristic. Other differences may include the induction of cardioplegia and hypothermia on the clinical side and the use of open reservoirs and roller pumps (versus closed reservoirs and centrifugal pumps) in ECLS. For rodent models in CPB research, Ballaux et al. [9], Berner et al. [10], Samarska et al. [11], Jungwirth, de Lange [12], and Umei [13] provide excellent reviews of relevant publications.

However, most of the identified ECLS animal models appear to use the same or similar miniaturized membrane oxygenators originally designed for CPB research. These devices are not optimal for prolonged experimental durations. The most distinguishing feature of chronically used oxygenators is the choice of the fiber used. Almost all identified small animal oxygenators use microporous polypropylene (PP) fibers. These fibers are not stable after a few hours of use, mainly due to plasma leakage through the fibers but also due to fouling and condensation on the gas side of the membranes. Hence, (sub-)chronic small animal models are currently not possible, while many of the complications of ECLS occur only after days or weeks after therapy initiation. This implies that it is often difficult or impossible to model the processes that lead to a particular complication. Historically, ECLS therapy gained importance in human care when PP was replaced as the membrane polymer by the hybrid polymethylpentene (PMP) membrane with a solid outer phase for solution-diffusion mass transfer (see Evseev et al. for details [14]). The same principle also applies to in vivo studies. Note that it should be technically possible to implement other fibers without redesign, at least in the self-made models.

Apart from the materials, the blood rheology within any oxygenator plays an important role in the initiation and propagation of hemostasis, thrombosis, hemolysis, and inflammatory response. In contrast to CPB, almost half of all fatal complications in ECLS originate from therapy-induced coagulation or bleeding disorders [15]. Shear rates, low-flow and stagnation zones, and washout characteristics of the oxygenator are defined by the flow path design. Most of the identified rodent-sized oxygenators are kept simple with an axial flow path along axially aligned hollow fibers, similar to conventional hemodialyzers, but with the blood flow on the outer fiber surface. None of the designs are scaled-down versions of the more sophisticated flow-optimized ECLS devices currently on the market. This may explain why all of the publications to date have focused rather on the clinical aspects of their study and less on oxygenator technology. Only one of the oxygenator performances was evaluated in vitro (see Fujii et al. in Table 1 [16], referring to the original work by Yamada et al. [17]), as is standard for full-size devices in clinical use, regulated in the DIN EN ISO 7199 [18].

All studies listed in the table are either preliminary or proof-of-concept efforts. The short experimental durations do not allow conclusions to be drawn about the long-term stability of the oxygenators. Design details are not consistently available: Two of the oxygenators are still commercially available (“Membrane oxygenator for small animal experiments”, Xi’an Xijing Medical Appliance Co., Ltd., Xi’an, Shaanxi, China [19] and “Gas Exchange Oxygenator, Miniature”, Living Systems Instrumentation, Catamount Research and Development, Inc., St. Albans, VT, USA [20]). For three other devices, no information on their commercial availability could be obtained (in Table 1, see “SAMO”, used by Fichter et al. [21], Warenits et al. [22], Magnet et al. [23] Edinger et al. [24,25]), as well as “Micro-1”, used by Chang et al. [26], Cho et al. [27], Fujii et al. [28], Edinger et al. [25,29]). One oxygenator is a miniaturized hemodialyzer (see Vu et al. [30] in Table 1). The remaining devices are in-house fabrications without details on design and manufacturing.

To overcome the technical limitations of current rodent oxygenators stated above, we developed the so-called “RatOx”, which is specifically designed to be customized for a variety of rodent models. In addition, several other design features have been incorporated to improve versatility and the ease of use and enable studies focusing on oxygenator technology alongside clinical research.

The oxygenator offers multiple degrees of freedom in sizing and hollow fiber properties. Its design is scaled down from state-of-the-art human care oxygenators. The novel oxygenator is scalable for different rodent species. It can be fabricated with a minimal amount of hollow fibers for experiments such as surface cellularization/endothelialization [31,32], or it can be fabricated for maximum efficacy as in full-support scenarios of larger rodents. Care has been taken to design an oxygenator that many groups can use and potentially even manufacture themselves. We intend to have the RatOx oxygenator used as a standard test platform, both in vitro and in vivo. This will allow data pooling and analyses across models and institutions.

**Table 1 micromachines-14-00800-t001:** Rodent ECLS models. Details on test setup and oxygenator specifications. ECC: Extracorporeal circuit; ECPR: Extracorporeal cardiopulmonary resuscitation; CPB: Cardiopulmonary bypass; VV-ECMO: Veno-venous extracorporeal membrane oxygenation; VA: Veno-arterial; AV: Arterio-venous; ECCO_2_R: Extracorporeal CO_2_ removal. SD: Sprague Dawley.

Authors, Year	Animal Model	ECLS Mode	Oxygenator Design and Membrane Type	Oxygenator Priming Volume	Effective Membrane Surface Area	Flow during Experiment	Duration of Experiment	Ref.
***Ali* et al. *2014***	SD-rat	ECPR	Undisclosed design/vendor. Images show axial intraluminal flow oxygenator. Silicone membrane.	8 mL	Undisclosed	5–6 mL/min. Flow at ECPR-induced arterial pressure of 25–30 mmHg.	7.3 ± 2.8 min ECPR + 30 min weaning	[33]
***Fichter* et al. *2016***	Fischer-344-rat	Ex vivo organ perfusion	Oxygenator named “Small Animal Micro Oxygenator” (SAMO). Three-layer stacked membrane mats of undisclosed size (5-10 cm edge length, approximated from published image).Polypropylene membranes.	10 mL for the entire ECC	Undisclosed	2 mL/min to perfuse isolated free flap	8 h	[21]
***Warenits* et al. *2016***	SD-rat	ECPR	“SAMO”-oxygenator, see Fichter et al. in this table. Membrane type undisclosed in this publication; group has worked with the same device with polypropylene fibers (see Magnet et al. and Fichter et al. in this table).	Undisclosed	Undisclosed	100 mL/kg/min	10 min + 43–83 min weaning	[22]
***Wiegmann* et al. *2016***	In vitro	N/A	In-house design. Single-fiber-mat design. Not for actual ECLS therapy but for experimental endothelialization of hollow-fiber surfaces (oxygenator-like flow chamber). Heparin/albumin-coated polymethylpentene fibers, experimental endothelialization.	4.275 mL	18.75 cm^2^ fiber mat area, i.e., 40 cm^2^ effective membrane surface area	15, 30, 60, 90 mL/min	96 h	[31]
***Magnet* et al. *2017***	SD-rat	ECPR, CBP features	“SAMO”-oxygenator, see Fichter et al. in this table. Polypropylene membrane.	Undisclosed, 15 mL for the entire ECC	Undisclosed	100 mL/kg/min with rats btw. 460 and 510 g	Max. 10 min + 30 min weaning	[23]
***Chang* et al. *2017***	Wistar–Kyoto rat	ECPR	“Micro-1” rat oxygenator (Dongguan Kewei Medical Instrument Co., Ltd., Guangdong, China). Current commercial availability unknow. Axial flow oxygenator with unknown intra- and extraluminal phases. Membrane type undisclosed in this publication; another group listed below (Cho2021) has likely used the same device from the same manufacturer with polypropylene.	Undisclosed, 19–20 mL for the entire ECC	Undisclosed; Cho et al., also in this table, have likely used the same device by the same vendor (500 cm^2^)	Undisclosed	30 min	[26]
***Madrahimov* et al. *2018***	C57BL/6 mouse	VV-ECMO	CPB-oxygenator, in-house design, axial intraluminal flow. Polypropylene membrane. Further information in other publications of Madrahimov et al. [34,35]	Undisclosed,<0.3 mL in referenced [34]	Undisclosed, 50 fibers of 80 mm each of undisclosed outer diameter	1.5–5 mL/min	2 h ECC + 5 min weaning	[36]
***Natanov* et al. *2019***	C57BL/6 mouse	VV-ECMO	CPB-oxygenator, in-house design, axial intraluminal flow. Polypropylene membrane. See Madrahimov et al. in this table for further information.	Undisclosed, 500µL for the entire ECC	Undisclosed, 50 fibers of 80 mm each of undisclosed outer diameter	Undisclosed, 3-5 mL/min in [36]	4 h	[37]
***Vu* et al. *2019***	SD-rat	ECCO_2_R/Dialysis	“M10” miniaturized dialyzer (Gambro, Lakewood, USA). Only previously described by the group in May et al. [38] and by Goldstein et al. [39]. Axial intraluminal flow oxygenator for neonates. Membrane so-called “AN69” hydrophilic hollow fiber made of s a copolymer of acrylonitrile and sodium methallyl sulfonate. See Thomas et al. [40] for details.	Undisclosed, but the device was designed for infants of 2–15 kg.	420 cm^2^	1 mL/min	Undisclosed	[30]
***Wollborn* et al. *2019***	SD-rat	VA-ECMO vs. ECPR	“OX “ miniature gas exchange oxygenator, Living Systems Instrumentation, St. Albans City, Vermont, USA. Not further described in the study. The manufacturer’s product description shows an axial extraluminal flow oxygenator and a polypropylene membrane [20].	Undisclosed, 6 mL for entire ECC. Product sheet by the vendor: 1.6 mL	Undisclosed. Supplier: 115 cm^2^ [20]	10–18 mL/min, to reach mean arterial pressure 65 mmHg	2.5 h + undisclosed weaning	[41]
***Fujii* et al. *2020***	SD-rat	VA-ECMO	“Micro-1” (Senko Medical Instrument Mfg. Co., Ltd., Tokyo, Japan). The picture shows an axial flow oxygenator, apparently with intraluminal flow. The oxygenator appears to be shorter but larger in diameter than other axial flow oxygenators in this table. The oxygenator could not be identified from the manufacturer’s product lists. Membrane type undisclosed.	Undisclosed, 8 mL for entire ECC	Undisclosed	70 mL/kg/min	2 h	[28]
***Edinger* et al. *2020***	Lewis rat	VA-ECMO	The “Micro-1” (see Chang et al. in this table) was tested against the “SAMO” (see Fichter et al. in this table) without further device specifications. Both membrane types undisclosed, both polypropylene in other publications with the same devices (Magnet et al. and Cho et al. in this table)	SAMO: 7 mLMicro-1: 3.5 mL	SAMO: 500 cm^2^; Micro-1: Published with 50 cm^2^ (Cho et al. published 500 cm^2^).	90 mL/kg/min	2 h	[25]
***Wilbs* et al. *2020***	New Zealand white rabbit	VV-ECMO	In-house design with 40 stacked fiber mat layers. Despite incorporating genuine hollow fibers, the oxygenator was built non-functional regarding gas transfer. It can be considered a simplified mock oxygenator for hemocompatibility testing. Polymethylpentene membrane. This fiber arrangement and fiber bundle design are similar to the RatOx-oxygenator. The stacked fiber mat layers have a cross-sectional flow area that is half that of the RatOx.	Undisclosed	263 cm^2^.	45 mL/min	4 h	[42]
***Li* et al. *2021***	SD-rat	VV-ECMO	CPB-oxygenator, axial extraluminal flow. Manufactured by Xi’an Xijing Medical Appliance Co. Limited, Xi’An, China. The membrane type is undisclosed in the publication, but based on an inquiry with the supplier, polypropylene fibers are used.	3 mL	200 cm^2^	80–90 mL/kg/min	3.5 h	[43]
***Cho* et al. *2021***	SD-rat	VA- vs. VV-ECMO	“Micro-1”-oxygenator; see Chang et al. in this table. Polypropylene membrane.	3.5 mL	500 cm^2^	50 mL/min	2 h	[27]
***Fujii* et al. *2021***	SD-rat	VV-ECMO	Axial, extraluminal flow oxygenator. Note that the figure in this publication suggests an intraluminal flow, whereas the original publication by Yamada et al. clearly indicates an extraluminal flow. This original publication also includes two other slightly larger oxygenator variations. They also state that polypropylene fibers are used. [17]	Undisclosed, 8 mL for entire ECC. Yamada et al. state a priming volume of 3 mL [17].	Undisclosed. Yamada et al.: 236 cm^2^ [17]	50–60 mL/kg/min	2 h	[16]
***Umei* et al. *2021***	SD-rat	Mock-ECLS, supported AV-ECMO	A 3D-printed flow cell designed to simulate the local geometry, blood flow patterns and surface area to blood volume ratio of a commercial oxygenator hollow-fiber bundle. Unable to transfer gas. The membrane type is non-functional, clear acrylate resin (PR-48, Colorado Polymer Solutions, Boulder, CO, USA).	0.3 mL for the oxygenator, 2.5 mL for the entire ECC	15 cm^2^	1.9 mL/min	8 h	[13]
***Edinger* et al. *2021***	Lewis rat	VA-ECMO	“SAMO”-oxygenator; see Chang et al. 2017 in this table. Polypropylene membrane.	Undisclosed, 11 mL for the entire ECC	Undisclosed, 500 cm^2^ in publication of Edinger et al. (2020) in this table	90 mL/min	2 h	[24]
***Govender* et al. *2022***	Syrian golden hamster	VA-ECMO	ECC-setup without oxygenator.	Not applicable	Not applicable	15% of CO	1.5 h ECC	[44]
***Greite* et al. *2022***	C57BL/6 mouse	VV-ECMO	Redesigned from [35]; CPB-oxygenator, in-house-design, axial intraluminal flow. Polypropylene membrane.	200 µL	Undisclosed, 50 fibers of 80 mm each of undisclosed outer diameter	Undisclosed, 2.34–6.5 mL/min in [35]	2 h ECC + 2 h weaning	[45]
***Huang* et al. *2022***	SD-rat	VV-ECMO	See details from Li et al. in this table.	See details from Li et al. in this table.	See details from Li et al. in this table.	See details from Li et al. in this table.	3 h	[46]
***Zhang* et al. *2022***	SD-rat	VV-ECMO	See details from Li et al. in this table.	See details from Li et al. in this table	See details from Li et al. in this table	80–90 mL/kg/min	2 h	[47]
***Edinger* et al.**	Lewis rat	VA-ECMO	“Micro-1 “-oxygenator; see Chang et al. 2017 in this table. Polypropylene membrane.	9 mL	Undisclosed, 500 cm^2^ in publication by Cho et al. in this table	90 mL/min	2 h	[29]
***Kharnaf* et al. *2022***	C57BL/6 mouse	VA-ECMO	“OX” miniature gas exchange oxygenator; see Wollborn et al. in this table.	1.6 mL oxygenator priming (NaCl, Hetastarch,), 2 mL in remaining ECC (allogeneic blood)	115 cm^2^ (supplementary Materials)	3–5 mL/min	1 h	[48]
***Alabdullh* et al. *2022***	In vitro	N/A	In-house design. Single-fiber-mat design. Not for actual ECLS therapy but for experimental endothelialization of hollow-fiber surfaces (oxygenator-like flow chamber). The oxygenator design is further development of the device previously published by Wiegmann et al. in this table. Heparin/albumin-coated polymethylpentene fibers with endothelialization.	4 mL	19 cm^2^	Static	6 h and 24 h	[32]

In this publication, we describe the novel rodent-dedicated oxygenator and preliminary results of in vitro classification experiments for the most relevant performance parameters, i.e., gas transfer and priming volume. To assess the transferability of the system to other institutions, we compare our initial results with those obtained by an independent laboratory that performed an additional test series using a different set of equipment but the same oxygenator design and methods.

## 2. Materials and Methods

### 2.1. Oxygenator Design

Based on the identified existing devices for rodent ECLS models and their applicability restrictions, we deducted and synthesized requirements for a novel system with the aim of overcoming as many limitations as possible. We further defined requirements that go beyond these limitations. A summary of all design targets is presented in Table 2.

The requirements for size and dimensions were derived from the physiology of laboratory animals ranging in size between mice and guinea pigs. These values were compiled amalgamated from various sources in the literature, including the studies listed in Table 1. All identified values are listed in Table 3 below. Since the majority of the studies in Table 1 used Sprague Dawley rats, a full-support ECLS scenario for a young rat was defined as the intended upper functionality boundary.

### 2.2. In Vitro Proof-of-Concept

Further, in vitro experiments to evaluate the performance of the RatOx oxygenator were performed. We adapted the test setups from our established protocols for full-size human oxygenators [61,62,63]. The latter are in accordance with the DIN EN ISO 7199 [18], which is part of the regulatory process for commercialized ECLS oxygenators.

All experiments were performed with fully heparinized (15.000 IU/L, Ratiopharm, Ulm, Germany) fresh porcine whole blood from a local slaughterhouse, which was immediately treated with 1.6 mL/L gentamicin (Ratiopharm, Ulm, Germany), 1.8 mL/L 50% glucose (B.Braun, Melsungen, Germany), and 100 mL/L 0.9% NaCl (B.Braun, Melsungen, Germany). The hemoglobin concentration was maintained at 12 +/− 1 mg/dL blood, the temperature at 37 +/− 1 °C, and the base excess at 0 +/− 5 mmol/L using 8.4% hydrogencarbonate (B.Braun, Melsungen, Germany). The test loop consisted of a blood pump, the oxygenators to be tested, a heat exchanger, ports for sampling before and after to the oxygenators, and sensors for pressure, flow, and temperature. A second circuit was used to obtain physiologic venous blood gas levels before each oxygenator, as defined in ISO 7199 (SO_2_ = 65 +/− 5%, p_CO2_ = 45 +/− 5 mmHg) [18]. Both circulations were connected to form a cross circuit using a self-built spillover hard-shell reservoir from which the test loop draws conditioned blood. The adherence of venous values in the conditioned blood was closely monitored. We recorded values for oxygen transfer, carbon dioxide elimination and pressure drop on the blood side for different numbers of fiber mats and different blood flows. An ABL Flex 825 (Radiometer GmbH, Willich, Germany) was used for measuring blood gases. To calculate oxygen and carbon dioxide transfer (V_O2_ and V_CO2_), we used the absolute gas concentration values for physically dissolved and hemoglobin-bound oxygen as well as physically dissolved and chemically bound carbon dioxide (ctO_2_ and ctCO_2_). Gas transfer values were calculated using venous and arterial samples drawn immediately before and after the oxygenators. Differently sized oxygenators (29, 42, and 55 fiber membrane layers) were tested over a range of blood flows through the oxygenator. To ascertain comparable experimental data, we chose to test the maximum number of fiber mat layers (55), ~75% fiber mat layers (42) and ~50% fiber mat layers (29) while maintaining an even distribution of the fiber mat layers (distance of 13) as well as a factor of max fiber mat layers over tested fiber mat layers of 2 or greater. The blood flow rates tested were 60, 80, 90, and 100 mL/min. The corresponding sweep gas flow was at a ratio of 2:1 (gas flow/blood flow) with pure oxygen. To test whether the oxygenator shows a linear correlation at higher flows, we also tested 500 mL/min for the 42-layer RatOx.

An additional test series was performed at an external collaborating institute, the Engineering Organ Support Technologies group at the University of Twente, Netherlands. The group used the same RatOx design and manufacturing process for their in vitro blood tests as the one used by the University Hospital RWTH Aachen, with minor modifications permitted. The aim of this knowledge transfer within the present study was to evaluate the feasibility of cross-institutional RatOx use. Apart from technical information, nothing else was provided, including parts, manufacturing tools, or materials. The University of Twente implemented and adapted the manufacturing modalities to their individual conditions. Specifically, they modified the potting process for the fiber bundle, thereby accelerating the potting process and building fiber bundles with 56 fiber mat layers, above the originally designed maximum of 55 layers. Further, they used Elastosil 625 silicone (Mc Technics, Visé, Belgium), which has stiffer mechanical properties than the Elastosil 620 used in Aachen. Most other manufacturing process modalities are similar and can be requested from our research groups. Ten identical fiber bundles were produced in this manner. Since the fibers are the most influencing factor of the oxygenator, the same uncoated PMP fibers were used (Membrana GmbH, 3M, Wuppertal, Germany). Apart from the oxygenator, the University of Twente implemented a test circuit also based on the DIN EN ISO 7199 [18]. In contrast to the procedure in Aachen, the blood gas analyzer was an i-STAT Alinity POC (Abbott, IL, USA), the heat exchanger heated the blood in the conditioning loop and not proximal to the test section, and the blood flows tested were 60, 100, 140 and 180 mL/min, with constant sweep gas of 500 mL/min.

Last, we tested several manufactured RatOx-devices using water for their priming volume in order to confirm the analytically derived Equation (1) (Section 3 below). We defined the internal volume of the RatOx as the entity of extraluminal fluid within all blood-leading parts and between the entry and exit ports to which the circuitry (blood tubing) is connected.

## 3. Results

### 3.1. Oxygenator Design

The RatOx oxygenator is a scaled-down version of a stacked-design oxygenator with round potting. The basic design, therefore, reflects the state-of-the-art [64] and is comparable, for example, to the Hemovent-oxygenator for adult patients [65]. Figure 1 shows an image of a fully assembled RatOx oxygenator manufactured in our laboratory. Figure 2 depicts its assembled components, sliced at the middle plane, placed on a stand: the cylindrical housing (dark transparent gray) holds the fiber bundle (white/blue), which is compressed by two disc-shaped lids (red) that also house the inlet and outlet Luer connectors for standard ¼ tubing. The compression force is provided by a large hand-operated screw (green) that moves relative to the housing caps (yellow). The fiber bundle, which can be adjusted in size by changing the number of fiber mat layers, is shown in Figure 3 in a medium size version with 29 layers.

The dimensioning of the RatOx oxygenator was based on the physiology of rats and related rodents (Table 3). The effective gas transfer was estimated based on available gas transfer data available in the literature (e.g., Evseev et al. [14]). The size of the oxygenator can be varied by the number of fiber mat layers, depending on the intended experimental model. It is also possible to select fiber module sizes for individual animals based on their body weight in a heterogeneously distributed animal cohort. The RatOx oxygenator allows for any number of fiber mats between 1 and 55, depending on the demand of the ECLS or animal model; for example, using standard polymethylpentene (PMP) fibers (Membrana GmbH, 3M, Wuppertal, Germany) with an outer diameter of 380 µm and a fiber pitch of 200 µm results in an effective gas exchange area between 10.6 cm^2^ and 580 cm^2^ (linear correlation). In the present study, all further calculations and tests were performed with these PMP fibers. The fiber mats are normally intermittently rotated by 90°, although technically, any angle between the fiber mats could be applied. The gas and blood sides of the hollow fibers are separated by a potting process using colored silicone (Elastosil RT 620, Wacker Chemie AG, Munich, Germany); the gas inlet and outlet are also separated by this potting. Figure 3 shows a single-fiber module with 29 fiber mat layers. After the round potting, the flow path through the fiber bundle has a diameter of 25 mm, while the height depends on the number of fiber mats.

The priming volume of an empty RatOx was determined from CAD drawings to be 1.0 mL, with each fiber mat layer adding 0.075 mL of priming volume (Equation (1)). The priming volume is calculated from the top edge of the top connector to the bottom edge of the bottom connector, i.e., all fluid compartments within the red and blue/white parts of the RatOx depicted in Figure 2. Fiber mat tolerances are neglected.

This priming volume affects the clinically relevant hemodilution (*Equation (2)*)) below:


*Equation (1*
*)—Estimation of the priming volume of a RatOx-oxygenator*

(1)
$$V_{Priming}=1.0\;mL+n_{fiber\;mat\;layers}\cdot 0.075\;mL$$




*Equation (2)*
*—Function to estimate maximum priming volume of the RatOx (V_priming_) without diluting the blood below a tolerable hematocrit (H_m_) for a given initial blood volume of the animal (V_B_) and a given initial hematocrit (H_i_).*

(2)
$$\left(\frac{H_{i}}{H_{m}}-1\right)*\;V_{B}=V_{Priming}$$



For hollow fibers, any commercially available fiber type can be used as long as the fibers are knitted into fiber mats. Oxygenator fibers are usually made of PMP for long-term applications or microporous polypropylene (PP) for short-term applications [14]. These fibers can be obtained as monolayer fiber mat spools (Membrana GmbH, 3M, Wuppertal, Germany). Alternatively, commercially available oxygenators can be cut open and the fibers scavenged. In this way, the post-production modifications to the fiber mats by the oxygenator manufacturer, such as antithrombotic coatings [14,66,67], can be transferred to the rodent model. For groups working with their own fibers, surface modifications, or coatings, any fiber can be used as long as they are knitted into mats [66,68,69,70].

For cost-effectiveness, the fiber bundle is the only component designed as disposable. Transparent polymethylmethacrylate (PMMA) was selected for the housing to allow for air-free priming and leak detection. Components in direct blood contact must further be hemocompatible, which is why we used PMMA or polycarbonate (PC). All reusable parts must be sterilizable. We chose polyoxymethylene (POM) as a material for components without blood contact and without the need for transparency (screw and outer caps). To reduce costs and to be able to analyze the fiber bundle post-experiment (see Figure 4), the RatOx housing can be non-destructively disassembled. All parts can be manufactured on a 3-axis machining center without the need for complex equipment.

### 3.2. In Vitro Proof-of-Concept

The priming volume calculated by *Equation (1*) was tested for random fabricated fiber modules, and the function (*Equation (1*)) could be confirmed for each device. To do this, we defined the internal volume of a RatOx as starting with the outer edge of the Luer connector on the blood-leading parts and stopping at the outer edge of the opposite connector. In Figure 2, this is the extraluminal space with all red and white components.

Further, the gas transfer efficacy was experimentally evaluated. Figure 5 shows the results. Oxygen transfer (positive values) and carbon dioxide transfer (negative values) are shown for four different oxygenator blood flows and three different fiber mat layer numbers (29, 42, 55 layers). Blood flows of 60, 80, 90, and 100 mL/min were evaluated. The zoomed-out graph shows the same results with an additional operating point of 500 mL/min for a 42-layer RatOx.

Using the same example as above, for a 55-layer module (*n* = 1) and a blood flow of 100 mL/min, the gas transfer of the RatOx oxygenator is approximately 5.53 mL/min for oxygen and −5.50 mL/min for carbon dioxide, respectively. The same 55-layer RatOx at a blood flow of 60 mL/min yielded an average oxygen transfer of 3.88 mL/min and a carbon dioxide transfer of −3.76 mL/min. The 29-layer RatOx (*n* = 4) showed values of 3.65 ± 0.28 mL/min oxygen transfer, −2.52 ± 1.05 mL/min carbon dioxide transfer (both at 100 mL/min) and 2.85 ± 0.32 mL/min oxygen transfer, −3.05 ± 0.39 mL/min carbon dioxide transfer (both at 60 mL/min), respectively.

Additional testing in a second, independent laboratory yielded the following results, as shown in Figure 6, augmented with the comparable experiments in the original institute (extracted from Figure 5):

For increasing blood flows, the oxygenators exhibit an almost linear correlation between oxygen and carbon dioxide transfer rates. The mean oxygen transfer rate is 4.30 mL/min at 60 mL/min (standard deviation ± 0.41 mL/min), 6.27 mL/min at 100 mL/min (standard deviation ± 0.98 mL/min), 8.83 mL/min at 140 mL/min (standard deviation ± 1.77 mL/min), and 9.33 mL/min at 180 mL/min (standard deviation ± 0.73 mL/min). The comparable values from Aachen at 60 mL/min and 100 mL/min are 3.88 mL/min and 5.53 mL/min, respectively. The mean carbon dioxide transfer rate is −4.87 mL/min at 60 mL/min (standard deviation ± 1.10 mL/min), −8.20 mL/min at 100 mL/min (standard deviation ± 1.97 mL/min), −8.64 mL/min at 140 mL/min (standard deviation ± 3.50 mL/min), and −10.28 mL/min at 180 mL/min (standard deviation ± 3.37 mL/min). The comparable values from Aachen at 60 mL/min and 100 mL/min are −3.76 mL/min and −5.50 mL/min, respectively. All values are listed in Table 4.

## 4. Discussion

Overall, based on the results above, the RatOx oxygenator is performing as expected. All values were within or exceeding the intended range, with a deficiency in the maximum gas transfer performance. In the following, the results are discussed in more detail, reflecting on the predefined design requirements in Table 2 and the experiment results.

*State-of-the-art design:* The oxygenator design is directly comparable to recently marketed devices such as the oxygenators by Nautilus (MC3 Cardiopulmonary, Dexter, MI, USA) and Hemovent (Hemovent, Aachen, Germany, recently acquired by Shanghai MicroPort Medical Group Co., Ltd., Shanghai, China). [64,65] *Interchangeable fiber type:* Any type of hollow fiber with any surface modification can be implemented with minor restrictions on freedom of choice, such as the requirement that the fibers are woven into mats. *Variable gas exchange area:* The RatOx comes in 55 different sizes with a minimum of 10 cm^2^ and a maximum of 580 cm^2^ for standard PMP fibers from Membrana (Membrana GmbH, 3M, Wuppertal, Germany). The University of Twente has shown that, with a modified fiber bundle potting process, even larger stacks with more than 55 layers are possible. *Low priming volume:* For larger rodents such as rats, the priming volume < 6 mL translates into hemodilutions of less than 25%, which is in compliance with the pre-defined requirements. For smaller animals, such as mice, the minimum static priming volume of 1 mL can easily result in a hemodilution of around 50%, which is clinically more difficult to handle, depending on the experimental setup. *Low pressure drop:* Unfortunately, we lack continuous data regarding the pressure drop across the RatOx oxygenator and cannot comment on the results. *Hemocompatibility:* Moreover, we have not yet been able to perform hemocompatibility tests and therefore cannot comment. However, care has been taken to use only materials in blood contact known to be hemocompatible. In addition, the flow path design has been adapted from optimized full-size ECLS devices, and we do not anticipate hemocompatibility issues. *Transparent housing:* It is possible to directly detect entrapped gas bubbles in the blood flow between the edges of the fiber bundle and the cylindrical housing. It is also possible to detect undesired leaks due to defective fibers and seals. *Reusable housing and removable fiber bundle:* Except for the fiber bundle, all components of the RatOx-oxygenator are designed for repeated use. The fiber bundle can be easily explanted post-experiment for further analyses, as shown in Figure 4.

*Effective gas transfer:* For a given operating point of 100 mL/min blood flow, the oxygen transfer was measured to be 5.53 mlO_2_/min. This is only about 70% of the intended full support of a 280 g rat (Table 3) with 55 layers and 100 mL/min blood flow. The results from Twente suggest a 10% higher gas transfer performance. The mean gas transfer rates at the given 100 mL/min blood flow for *n* = 10 56-layer oxygenators showed values of 6.27 mL/min. The determined demand of 7.90 mL/min as a design requirement specification was based on a single publication (Bedford et al. from 1979 [49]) for awake, resting rats. The actual oxygen demand may be higher or lower. Choice of animal, degree of support, gas flow, vasoactive drugs, sedation, hypothermia, animal activity state, autologous and allogeneic blood transfusions, and the use of residual circulatory/pulmonary function are only a few options to compensate for the presumed lack of gas transfer performance. Merely increasing the effective surface area will also cause a higher priming volume, resulting in a conflict in objectives, i.e., a decrease in the tolerated hemodilution. Apart from the full-support ECLS mode, partial-support therapies will work well with the introduced system. This is why and how even larger animals, e.g., rabbits, can be used with the RatOx system. Wilbs et al. [42], also listed in Table 1, provided a corresponding example where they treated New Zealand White rabbits with a systemic anticoagulant and monitored the hemocompatibility of a non-functional ECLS unit.

We also tested two 42-layer oxygenators with an operating point well outside the relevant range for rodents. In general, the oxygen uptake for a given batch of blood increases with a slower blood flow through the oxygenator because there is more time for mass transport. This correlation is not valid for very low blood flows as the oxygen saturation reaches its limit before the blood has passed the later hollow fiber layers. A graph of arterial oxygen saturation over blood flow would start with a plateau at nearly 100%. It would then show a characteristic salient point at a certain blood flow from which the curve decreases linearly. To test the oxygenator for this expected linear correlation for higher blood flows, we added the operating point of 500 mL/min blood on the experiment day, where we also tested with the 42-layer oxygenator. This is far from the normal cardiac output of rodents and thus outside the range of operating points that can be run in an in vivo experiment, but the functional limit of the oxygenator for in vitro testing may be much higher. We aimed to demonstrate that any required gas transfer efficacy up to an operating point of 500 mL/min can be achieved.

*Reproducibility, cross-lab usability and manufacturability:* The experiment series at Twente University was conducted to identify potentials and limitations to the transfer of laboratory-grade test oxygenators between institutions. Despite major barriers to the technology transfer, such as local conditions, and equipment, the obtained test results provide an opportunity for discussion. The technology transfer was sufficiently successful in conducting standardized experiments according to DIN EN ISO 7199, despite a comprehensive manufacturing process. Both the oxygen and the carbon dioxide transfer results of the devices from Twente showed a higher efficacy. In the case of the oxygen transfer, the differences were minor; the results from Aachen lie within the range of the standard deviation from Twente. For the carbon dioxide transfer, the same is true only for the lower operating point of 60 mL/min blood flow, but the tendency for a larger discrepancy is visible for higher flows. As we only tested a single device in Aachen for a proof-of-concept, a statistical evaluation would be inconclusive. However, the potential differences between the test results from Aachen and Twente are plausible:

First, the standardized test methodology in DIN EN ISO 7199 leaves room for individual adjustments and adaptations. Blood handling, blood age, transport modalities after blood donation, blood collection procedures, etc., may differ. Second, the experiment setup and procedure may differ, e.g., in the duration between sample collection and blood gas measurement, in the test circuit design, or in the priming process. In this case, the heat exchanger was placed at two different locations with an unmonitored effect on the gas uptake/elimination in the test devices. This displays a particularity of the RatOx-system, as it does not contain heat-exchanger fibers, such as those in conventional oxygenators. Third, the sweep gas flow rate was adjusted to always be at a gas/blood flow ratio of 2:1 in Aachen but was kept constant over all operating points in Twente. While we know from the clinical experience with ECLS that the oxygen uptake is mainly dependent on the blood flow, the carbon dioxide elimination is mainly dependent on the sweep gas flow rate or gas/blood flow ratio [71]. In our results, the carbon dioxide transfer efficacies do indeed diverge stronger than those for oxygen.

Apart from the test circuit, the device itself may also cause differences in the results. The University of Twente has made adjustments and improvements to the manufacturing process that may have a (positive) effect on the gas transfer efficacy. For example, Twente used a more rigid silicone, which may have resulted in improved fiber patency following the cutting process to reopen the fibers after potting or as a counterforce to the compression after the device assembly. Additionally, one additional fiber mat was included in the devices of Twente, increasing the functional gas transfer surface area by ~2%; on the other hand, the average flow path cross-sectional area was 1 mm smaller, effectively reducing the functional gas transfer surface area of the fiber bundle by 4%. Both effects should partially neutralize each other.

## 5. Conclusions and Outlook

This publication introduces a novel dedicated rodent oxygenator. We have performed preliminary experiments to prove our concept. Overall, we conclude that the RatOx oxygenator is a potential new tool for advanced small animal in vivo testing as well as accelerated in vitro testing of artificial lung support.

The RatOx oxygenator is designed to be a highly versatile, reproducible oxygenator that aims to become a standard tool in ECLS research and development. The defined requirements reflect this goal. Our preliminary laboratory results are promising to allow for various different experimental setups, both in vivo and in vitro. The oxygenator is extremely versatile, as the fibers can be selected with minimal restrictions. Figure 5 and Figure 6, (Equation (1) and Equation (2)), can be used to estimate the required number of fiber mat layers for any animal model design based on the required gas transfer performance demand and/or the maximum tolerable hemodilution. Physiological values for relevant animals are listed in Table 3. Further, fibers can be functionalized through modifications and coatings. The adaptable size of the oxygenator allows for different models, such as different rodents, different ELCS support types, or even low-surface-area endothelialization. At the same time, the oxygenator is very similar in design to full-size, commercially available oxygenators, allowing for direct comparisons.

We conclude that the RatOx oxygenator may enable research and development at much earlier technological stages and with higher throughput. This opens the path for radically new experimental setups and models, improving the scientific output for ECLS and, in the long run, the therapeutic outcome for patients suffering from severe sicknesses.

In addition to in vivo studies in rodents, a highly miniaturized ECLS system also allows for increased-throughput in vitro testing. Due to the small priming volume, several test circuits can be used in parallel, and human blood donations suffice for the experiments. This allows parameter studies on, e.g., design features, material properties, hemocompatibility and more.

Because an ECLS system can be used to bridge cardiorespiratory failure, it can be used to keep an entire organism perfused, effectively limiting necrotic and apoptotic processes and allowing for a near-physiological metabolism. With a rodent-sized ECLS system, euthanized animals can be used for studies that require any form of metabolism, yet without keeping the animal alive. In this way, animal trials can be reduced with positive ethical and economic implications.

We now aim to establish the RatOx in further in vitro setups, e.g., hemocompatibility testing, as well as in further animal models. The main limitation of the presented experimental methods is the low number of repetitions, as we presented preliminary results. Furthermore, we did not continuously measure the pressure drop, which is part of the standardized classification protocol. Our future goal is, on the one hand, to provide comprehensive in vitro and in silico classification experiments and, on the other hand, to collect RatOx-specific technical data from collaborating institutes from their experiments with the RatOx-system.

To complement the system, we are further developing the remaining components of a rodent-sized ECMO circuit (pump, reservoir, heat exchanger) to the same specifications.

We hope for many groups to adapt the RatOx platform to establish a database of results, enabling cross-laboratory and broader, collaborative research. The results from our experiments in Twente are very promising and show that a transfer of the technology to other institutions is feasible. Potentially, these very promising results are difficult to repeat for institutions with less experience and equipment for medical device manufacturing compared to the University of Twente; this problem can be overcome to a large extent by providing not only technological information between laboratories but also actual equipment. Since many groups have already used various other oxygenators, the handling of the circuit, apart from the manufacturing process, is well established in many laboratories. This is especially true if groups primarily involved in CPB research are included.

## Figures and Tables

**Figure 1 micromachines-14-00800-f001:**
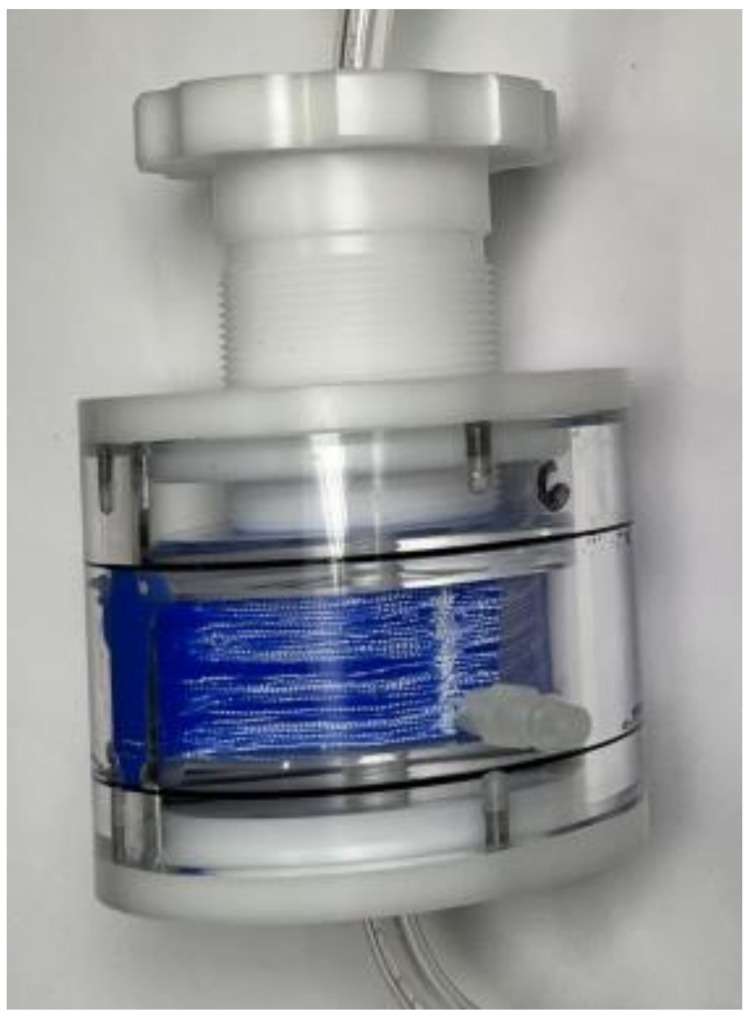
Fully assembled, maximum size RatOx prior to an in vitro experiment. Blue-white: Silicone potted fiber bundle (55 layers).

**Figure 2 micromachines-14-00800-f002:**
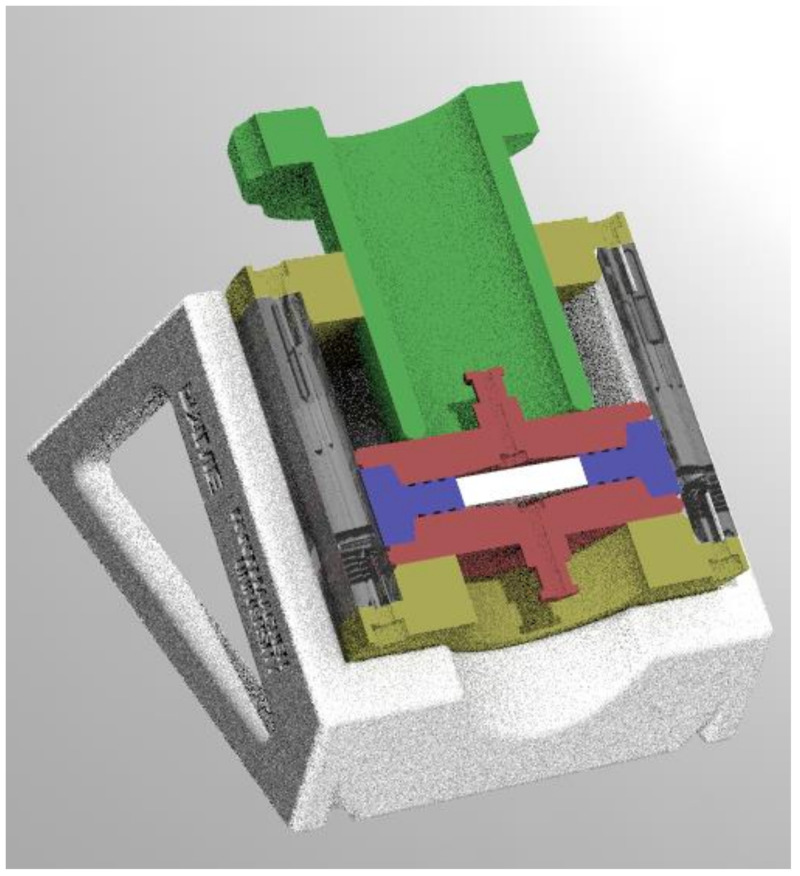
CAD sketch of a RatOx oxygenator on a pedestal. White: Size-variable fiber mat; blue: Silicone potted around the fiber mat, forming a fiber bundle; red: Top and bottom cap with connector for circuit tubing; dark gray: cylindrical transparent housing; yellow: Outer caps; green: Screw to secure all components with varying fiber module heights.

**Figure 3 micromachines-14-00800-f003:**
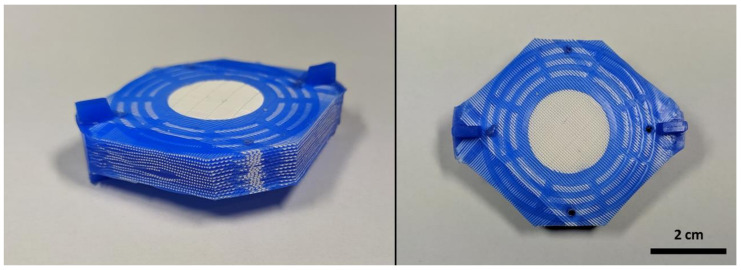
Photograph of a mid-size, silicone potted fiber module (29 layers), re-opened fibers, prior to assembly.

**Figure 4 micromachines-14-00800-f004:**
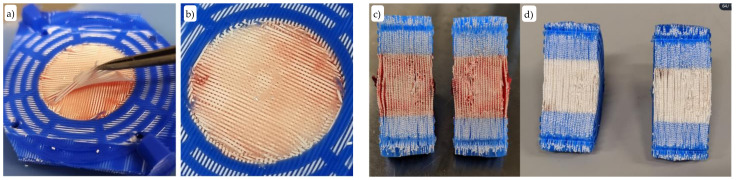
Examples of fiber modules explanted post-experiment for visual inspection. (**a**,**b**) Fiber bundle cut layer by layer for visual inspection. (**c**) 55-layer fiber module with uncoated PMP fiber mats from a disassembled RatOx oxygenator after an experiment with low-anticoagulated, fresh, whole, porcine blood. The module was cut in a middle plane. (**d**) An identical module with blood from the same donor in the same experiment but with high anticoagulation.

**Figure 5 micromachines-14-00800-f005:**
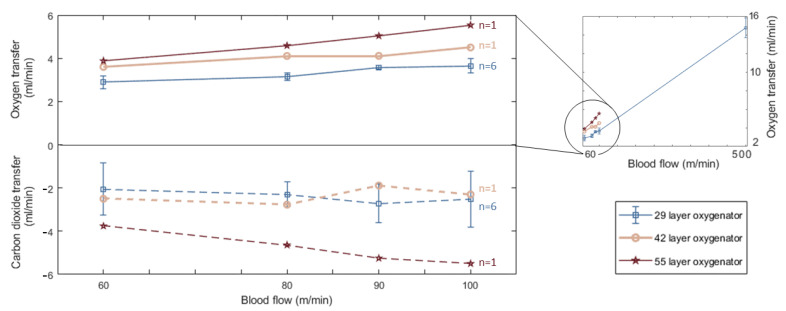
Gas transfer efficacy over blood flow for three different fiber mat layer numbers. The zoomed-out graph on the right further shows two measured values for 500 mL/min blood flow to show that a linear extrapolation of the values measured in the lower flow regimes is approximately valid. Blue squares: 29 fiber mat layers, beige circles: 42 fiber mat layers, red stars: 55 fiber mat layers. Solid lines: Oxygen transfer performance, broken line: carbon dioxide transfer performance.

**Figure 6 micromachines-14-00800-f006:**
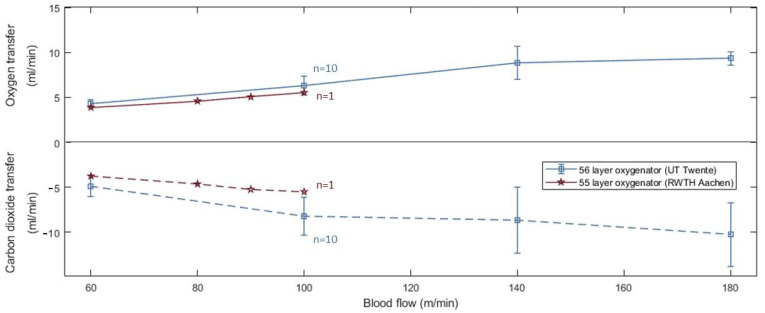
Oxygen and carbon dioxide transfer rates over blood flow. Blue: *n* = 10 experiments with 56 fiber layer oxygenators manufactured and tested at Twente University. Red stars: Similar experiments with a single 55-*layer* mat oxygenator, manufactured and tested at the University Hospital Aachen (identical data also previously shown in Figure 5). Solid lines: Oxygen transfer performance, broken line: Carbon dioxide transfer performance.

**Table 2 micromachines-14-00800-t002:** Requirement specifications and design targets.

Requirement	Design Target
**State of the art design**	The oxygenator is built using state-of-the-art design principles. The basic functions are scaled down from commercially available oxygenators.
**Interchangeable fiber type**	All available and similar hollow-fiber membrane types can be used.
**Effective gas transfer**	Based on a rodent body weight of 280 g and a (resting) O_2_ demand of 0.028 mLO_2_ per gram body weight per minute (Sprague Dawley rats, both parameters published by Bedfort et al. already in 1979 [49]), the maximum oxygen transfer capacity of the rodent oxygenator should be at least 7.9 mLO_2_/min. This is valid for blood flow values similar to the physiological cardiac output (121 mL/min, for a 350 g Sprague Dawley rat, published by Brands et al. [50] translates to approximately 100 mL/min for a 280 g rat, assuming the validity of a linear interpolation).
**Variable gas exchange surface**	The design of the oxygenator allows the use of differently sized fiber modules for different animal models and ECLS modalities. The surface area is between 10 cm^2^ for experimental incubation with scarce endothelial cells (25% of the surface area seeded in a specially designed incubator by Wiegmann et al. [31]) and sufficient gas transfer exchange area toallow the targeted 7.9 mLO_2_/min of transfer.
**Low priming volume**	The static priming volume is kept as low as possible to minimize hemodilution. The gas exchange area is adjustable to keep the variable priming volume low (see above). The blood volume of a 280 g Sprague Dawley rat can be estimated to be 20.44 mL [51]. Depending on the model, a hemodilution of 20–30% can be tolerated, resulting in a maximum priming volume of approximately 5–9 mL.
**Low pressure loss**	To avoid the need for large pumps and to allow for arteriovenous ECLS cannulation, the pressure drop due to flow resistance across the oxygenator does not exceed 25% of the average mean arterial pressure of the animal, analogous to full-size oxygenators in humans. In the Sprague Dawley rat, this means a tolerable pressure drop of 25 mmHg.
**Hemocompatibility**	Blood-leading components of the oxygenator do not cause avoidable hemocompatibility issues and undesired influences in hemocompatibility testing. Since there are no reference values for oxygenators for hemocompatibility to test this design target, especially not for rodent-sized oxygenators, only state-of-the-art oxygenator designs and materials are used.
**Transparent housing**	The oxygenator is designed to allow for the detection of air bubbles and plasma leakage, using transparent materials where necessary.
**Reusable housing and removable fiber bundle**	To reduce economic burden and to increase availability, the oxygenator design allows for non-destructive disassembly, e.g., without adhesives. All blood-leading materials are made of sterilizable materials. The (disposable) fiber bundle can be explanted and mechanically opened for visual inspection (e.g., immunofluorescence, microscopy).
**Reproducibility, cross-lab usability and manufacturability**	The oxygenator fabrication and assembly process is simple and results in highly reproducible oxygenators. No special equipment is required to fabricate and deploy the oxygenator. The fabrication process uses low-tech equipment so that other laboratories and research groups can fabricate test objects on their own. If other groups do not possess sufficient equipment or resources, the oxygenator can be produced by collaborating institutions. This allows for the reproducibility of results data interpretation across studies, despite the large influence of the oxygenator unit on physiological systems such as hemostasis/thrombosis or inflammatory response [8].

**Table 3 micromachines-14-00800-t003:** Small animals potentially relevant for ECLS modeling with the RatOx oxygenator. The values in this table have been amalgamated both from published studies using the respective animal model (mostly for ECLS or CPB) as well as from textbook reference tables to fill in missing data. The data do not differentiate between sex, age, maturity status, pathology, feeding status, and, in some correspondingly denoted cases, strain. For further reading, excellent overviews of the individual animal were published by Fox et al. [52] and Suckow et al. [53]. Hgb: hemoglobin; Hct: hematocrit.

Rodent	Bodyweight (g)	Blood Volume (mL/kg)	Hgb (g/dL)	Hct (%)	Gas Demand O_2_(l/kg/h)	Heart Rate (min^−1^)	Mean Art. Pressure (mmHg)	Cardiac Index (mL/min/kg)	References	Comment
** *Mouse (C3H/HeJ, C57BL/6)* **	30 ± 5	80 ± 4	14 ± 2 ^(a)^	45 ± 7 ^(a)^	3.5 ± 1.5 ^(b)^	652 ± 25	92 ± 3	591 ± 49	[35,36,37,53,54,55]	^(a)^ Deer mouse ^(b)^ Approximation for 6 mouse species based on data from [55].
** *Gerbil* **	89 ± 43	73 ± 12	14 ± 4	44 ± 8	N/A	430 ± 170	89 ± 11	N/A	[53,56]	
** *Golden Syrian Hamster* **	100 ± 40	73 ± 7 ^(c)^	15 ± 5 ^(d)^	45 ± 15 ^(d)^	2.2 ± 0.9 ^(e)^	390 ± 110	113 ± 12	197.0 ± 19	[44,53,57,58]	^(c)^ Listed in [56] without specified hamster strain^(d)^ Listed in [53] without specified hamster strain ^(e)^ Gas demand decreasing distinctly with age (11–70 days)
** *Sprague Dawley rat* **	410 ± 190	58 ± 2	15 ± 2 ^(f)^	50 ± 3 ^(g)^	1.7 ± 0.1 ^(h)^	378 ± 64	105 ± 20	345 ± 20 ^(i)^	[22,43,49,51,53,54,59]	^(f)^ Values for rat strains kangaroo rat and cotton rat, from [53] ^(g)^ Values for rat strains kangaroo rat, listed in [53] ^(h)^ Calculated from values for animals of 280 g bodyweight ^(i)^ Calculated from values for animals of 350 g bodyweight
** *Chinchilla* **	500 ± 100	57 ± 24 ^(j)^	12 ± 3	43 ± 12	0.7 ± 0.1	125 ± 25	N/A	N/A	[53,56,60]	^(j)^ Estimated from absolute values given in [53]
** *Guinea Pig* **	950 ± 250	80 ± 13	14 ± 3	40 ± 10	0.8 ± 0.04	395 ± 75	67 ± 3	270 ± 30	[52,53,56]	

**Table 4 micromachines-14-00800-t004:** Oxygen and carbon dioxide transfer rates over blood flow for *n* = 1 experiments with 55-layer oxygenators conducted at the University Hospital Aachen and *n* = 10 experiments with 56 fiber layer oxygenators conducted at the Twente University.

Blood Flow (mL/min)	Aachen (55 Layers)		Twente (56 Layers)	
	Oxygen Transfer (mL/min)	Carbon Dioxide Transfer (mL/min)	Oxygen Transfer (mL/min)	Carbon Dioxide Transfer (mL/min)
** *60* **	3.88	−3.76	4.30 ± 0.41	−4.97 ± 1.10
** *80* **	4.59	−4.64	N/A	N/A
** *90* **	5.04	−5.25	N/A	N/A
** *100* **	5.53	−5.50	6.27 ± 0.98	−8.20 ± 1.97
** *140* **	N/A	N/A	8.83 ± 1.77	−8.64 ± 3.50
** *180* **	N/A	N/A	9.33 ± 0.73	−10.28 ± 3.37

## Data Availability

Please contact the authors for data and designs of the RatOx oxygenator.

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
