# Peer review of "Novel Size-Variable Dedicated Rodent Oxygenator for ECLS Animal Models—Introduction of the “RatOx” Oxygenator and Preliminary In Vitro Results"

_micromachines, 2023, doi:10.3390/mi14040800_

Round 1

Reviewer 1 Report

An exciting manuscript by German authors regarding the novel size-variable dedicated rodent oxygenator for ECLS animal models. I don't have any significant comments, just some minor comments to the authors:

1. The introduction seems too long since this is a research manuscript. Maybe the authors will transfer the table to supplementary materials.

2. The rest of the manuscript is very clearly and interestingly written - I have no criticisms here.

3. Author contributions must be completed.

Reviewer 2 Report

I am really thankful for the opportunity of reviewing this promising in-vitro validation of a novel rodent oxygenator device for ECLS.

In this field, we constantly encounter the issue of developing a valuable, reproducible, and, most importantly, standardized extracorporeal circuit and related components to investigate the effects of CPB, ECMO or ECLS.

As a proof of concept and preliminary validation study, the authors adopted an adequate methodology which sustains their results.

I have some minor comments especially regarding the clinical background and the discussion of findings:

-abstract: pls check grammar

-Intro line 40. ECLS: to date we cannot consider anymore ECLS as a last-resort measure since it has become the standard therapy for many (of course severe) conditions. Moreover, mortality needs to be contestualized

-Intro line 52-54: same as above. I disagree in putting together all kinds of ECLS and giving a survival number since it is really misleading

-Intro line 64: same as above

-Intro line 78-81: I suggest the authors to moderate this statement since the referred company may not share the authors' point of view on their "monopoly"

-Intro line 122-123: this assumption is not supported by evidence. 

-Intro line 123-133: pls uniform the refs captions style

-Table 2: if intended for ECLS, long-term biocompatibility and sustained performance should be considered too

-Results: what is the rationale of using 29, 42 and 55 layers specifically?

-Results: why if with 55 layers the authors achieved the 70% of the theoretic maximal support they didnt reformulate the oxygenator design to achieve full-support?

-Authors contributions part is missing

I suggest a general grammar/spelling check
